# Genomic and Molecular Profiling of Human Papillomavirus Associated Head and Neck Squamous Cell Carcinoma Treated with Immune Checkpoint Blockade Compared to Survival Outcomes

**DOI:** 10.3390/cancers13246309

**Published:** 2021-12-16

**Authors:** Hira Shaikh, Julie E. McGrath, Brittany Hughes, Joanne Xiu, Pavel Brodskiy, Ammar Sukari, Sourat Darabi, Chukwuemeka Ikpeazu, Chadi Nabhan, Wolfgang Michael Korn, Trisha M. Wise-Draper

**Affiliations:** 1Division of Hematology/Oncology, Department of Internal Medicine, University of Cincinnati, Cincinnati, OH 45267, USA; shaikhhl@ucmail.uc.edu; 2Department of Medical Affairs, Caris Life Sciences, Irving, TX 85040, USA; jmcgrath@carisls.com (J.E.M.); jxiu@carisls.com (J.X.); wmkorn@carisls.com (W.M.K.); 3University of Cincinnati Cancer Center, Cincinnati, OH 45267, USA; hughesbu@ucmail.uc.edu; 4R&D Lab Services, Caris Life Sciences, Irving, TX 75039, USA; pbrodskiy@carisls.com; 5Department of Oncology, Karmanos Cancer Institute, Wayne State University, Detroit, MI 48201, USA; sukaria@karmanos.org; 6Precision Medicine Program, Hoag Family Cancer Institute, Newport Beach, CA 92663, USA; sourat.darabi@hoag.org; 7Division of Medical Oncology, Department of Internal Medicine, University of Miami Miller School of Medicine, Miami, FL 33136, USA; cikpeazu@med.miami.edu; 8University of Miami Sylvester Comprehensive Cancer Center, Plantation, FL 33324, USA; 9Department of Precision Oncology Alliance, Caris Life Sciences, Irving, TX 75039, USA; cnabhan@carisls.com

**Keywords:** head and neck squamous cell carcinoma, oropharyngeal cancer, p16, human papillomavirus (HPV), personalized medicine, immune checkpoint blockade, outcomes

## Abstract

**Simple Summary:**

The prognosis of recurrent and/or metastatic (R/M) head and neck squamous cell carcinoma (HNSCC) remains poor. However, human papillomavirus (HPV)-associated oropharyngeal squamous cell carcinoma (OPSCC) patients live longer than those that are negative for HPV infection. In addition, some R/M HNSCC patients respond well to immune checkpoint blockade (ICB) therapies including pembrolizumab and nivolumab, but whether HPV infection is correlated with a good response to ICB is unclear. Here we attempt to understand if ICB treatment improves survival outcomes of HPV and/or surrogate marker p16−positive OPSCC and non-OP HNSCC. We also investigate other potential biomarkers and mutations that may predict improved response to ICB in both HPV−positive and -negative HNSCC patients. With better biomarkers, future treatment can be better tailored to individual patients to improve survival.

**Abstract:**

Recurrent/metastatic (R/M) head and neck squamous cell carcinoma (HNSCC) patients overall have a poor prognosis. However, human papillomavirus (HPV)-associated R/M oropharyngeal squamous cell carcinoma (OPSCC) is associated with a better prognosis compared to HPV−negative disease. Immune checkpoint blockade (ICB) is the standard of care for R/M HNSCC. However, whether HPV and its surrogate marker, p16, portend an improved response to ICB remains controversial. We queried the Caris Life Sciences CODEai database for p16+ and p16− HNSCC patients using p16 as a surrogate for HPV. A total of 2905 HNSCC (OPSCC, *n* = 948) cases were identified. Of those tested for both HPV directly and p16, 32% (251/791) were p16+ and 28% (91/326) were HPV+. The most common mutation in the OPSCC cohort was *TP53* (33%), followed by *PIK3CA* (17%) and *KMT2D* (10.6%). *TP53* mutations were more common in p16− (49%) versus the p16+ group (10%, *p* < 0.0005). Real-world overall survival (rwOS) was longer in p16+ compared to p16− OPSCC patients, 33.3 vs. 19.1 months (HR = 0.597, *p* = 0.001), as well as non-oropharyngeal (non-OP) HNSCC patients (34 vs. 17 months, HR 0.551, *p* = 0.0001). There was no difference in the time on treatment (TOT) (4.2 vs. 2.8 months, HR 0.796, *p* = 0.221) in ICB-treated p16+ vs. p16− OPSCC groups. However, p16+ non-OP HNSCC patients treated with ICB had higher TOT compared to the p16− group (4.3 vs. 3.3 months, HR 0.632, *p* = 0.016), suggesting that p16 may be used as a prognostic biomarker in non-OP HNSCC, and further investigation through prospective clinical trials is warranted.

## 1. Introduction

Head and neck squamous cell carcinoma (HNSCC) remains the sixth most common cancer worldwide despite recent advances in management, with more than 650,000 cases and 330,000 deaths annually [1]. It is predicted that by the year 2040, the worldwide incidence of HNSCC will increase by 32% and mortality by 34% [2]. Oral cavity, larynx, and hypopharynx cancers are often related to tobacco and alcohol. In contrast, most human-papillomavirus-positive (HPV+) cancers arise from the oropharynx (OP) [3,4]. HPV+ cancers usually demonstrate high p16 expression by immunohistochemistry (IHC), allowing p16 to serve as a surrogate for HPV infection [5,6]. Worldwide, approximately 25% of all HNSCCs are thought to be related to HPV [7]. In the Western world, including the United States and Europe, the incidence of HPV−associated HNSCC has risen substantially, while tobacco- and alcohol-related HNSCC has declined [4,8,9].

Previous studies have consistently shown that HPV−mediated (p16+) oropharyngeal squamous cell cancer (OPSCC) patients have better outcomes [10]. A retrospective analysis of HNSCC patients from the RTOG-0129 study, in which patients received definitive radiotherapy, demonstrated a three-year survival of 82.4% for p16+ compared to 57.1% in p16− patients [10]. The same was true in recurrent/metastatic (R/M) HNSCC. [11,12,13] To account for disparate outcomes between HPV+ and HPV− disease, genomic signatures have been previously explored [14]. For instance, *PIK3CA* mutations are common in HPV+ cancers, while *TP53* mutations are rare [15].

Despite improved outcomes for HPV+ disease, even in the R/M setting, the prognosis remains poor. Immune checkpoint blockade (ICB) is now approved for R/M HNSCC in the first line. However, HPV status and its role in prognosis remains unclear in R/M OPSCC upon treatment with ICB. Although not statistically significant, the overall response rate (ORR) was higher in HPV+ patients (24%) compared to HPV− patients (16%) in the KEYNOTE-012 study, which evaluated pembrolizumab in advanced, heavily pretreated HNSCC [16]. In the KEYNOTE-055 study, evaluating pembrolizumab in R/M platinum and cetuximab refractory HNSCC patients, no difference in progression-free survival (PFS) was observed between HPV+ and HPV− patients [17]. Additionally, in the CheckMate 141 trial, in which nivolumab was compared to the standard of care (SOC) in patients with platinum refractory, R/M HNSCC, a post hoc analysis demonstrated that HPV status did not confer a difference in outcome [18,19].

Given the inconsistent results, interrogation of biomarkers and genomic alterations is important to determine prognosis and potentially guide treatment paradigms in the future. Previously, high tumor mutational burden (TMB) and tumor immune infiltrate due to mutagen exposure has resulted in higher responses to immunotherapy [20,21,22]. In addition, HPV−related HNSCC has been shown to be enriched with tumor CD8 lymphocytes; the latter has been correlated with better outcomes with the use of ICB [23,24,25]. However, the significance of molecular, transcriptional, and immune signatures and the correlation with p16 expression and subsequent survival remains unclear.

Based on the above controversial data and increased projection of mortality rates, it is important to further elucidate the role of p16 and HPV in the outcomes of HNSCC patients who receive ICB. In addition, understanding molecular and transcriptional signatures in p16+ vs. p16− patients may indicate predictors of response that may better explain the characteristics of tumors likely to respond or be resistant to ICB to further guide treatment in the future. 

## 2. Methods

### 2.1. Samples

We queried the Caris Life Sciences database for p16+ and p16− HNSCC patients. Patients were considered smokers if they had >15 pack-years of tobacco use. Comprehensive molecular profiling, including whole-exome sequencing (WES), targeted Next-Generation Sequencing (NGS), whole transcriptome sequencing (WTS), and immunohistochemistry (IHC), was performed (Caris Life Sciences, Phoenix, AZ, USA).

### 2.2. Immunohistochemistry Analysis

p16 was determined by IHC, and a standard cut-off of 2+, >70% p16 staining was considered p16+. PD-L1 expression was assessed by the 22c3 antibody with a combined positivity score (CPS) of ≥1 being positive. CPS was determined by calculating the percentage of PD-L1-positive tumor cells, lymphocytes, and macrophages within the total number of viable cells. Mismatch repair (MMR) protein expression was tested by IHC using antibody clones (MLH1, M1 antibody; MSH2, G2191129 antibody; MSH6, 44 antibody [26]; and PMS2, EPR3947 (Ventana Medical Systems, Inc., Tucson, AZ, USA). The complete absence of protein expression of any of the 4 proteins (0+ in 100% of cells) tested was considered deficient MMR (dMMR). Microsatellite Instability Status (dMMR/MSI-H) was determined by a combination of multiple platforms to measure the MSI of MMR status of the tumor profiled, including fragment analysis (FA, Promega, Madison, WI, USA), IHC, and NGS.

### 2.3. Next-Generation Sequencing (NGS)

NGS was performed on genomic DNA isolated from formalin-fixed paraffin-embedded (FFPE) tumor samples using the nextSeq platform (Illumina, Inc., San Diego, CA, USA) at the Caris Life Sciences laboratory (Phoenix, AZ, USA). A custom-designed SureSelect XT assay was used to enrich 592 whole-gene targets (Agilent Technologies, Santa Clara, CA, USA). All variants were detected with >99% confidence based on allele frequency and amplicon coverage, with an average sequencing depth of coverage >500 and an analytic sensitivity of 5%. Prior to molecular testing, tumor enrichment was achieved by harvesting targeted tissue using manual microdissection techniques. Genetic variants identified were interpreted by board-certified molecular geneticists and categorized as ‘pathogenic,’ likely pathogenic,’ ‘variant of unknown significance,’ ‘likely benign,’ or ‘benign’ according to the American College of Medical Genetics and Genomics (ACMG) standards. When assessing mutation frequencies of individual genes, ‘pathogenic’ and likely pathogenic’ were counted as mutations while ‘benign,’ ‘likely benign,’ and ‘variants of unknown significance’ were excluded. Tumor mutational burden (TMB) was measured by counting all non-synonymous missense, nonsense, in-frame insertion/deletion, and frameshift mutations found per tumor that had not been previously described as germline alterations in dbsSNP151 and the Genome Aggregation Database (gnomAD) or as benign variants identified by Caris geneticists. The cutoff point of ≥10 mutations per MB was used based on the KEYNOTE-158 pembrolizumab trial.

### 2.4. Whole-Exome Sequencing (WES)

Direct Sequence analysis was performed on genomic DNA isolated from a microdissected, formalin-fixed, paraffin-embedded tumor sample using the Illumina Novaseq 6000 sequencers. A hybrid pull-down of baits designed to enrich for more than 700 clinically relevant genes at high coverage and high read-depth was used, along with another panel designed to enrich for an additional >20,000 genes at a lower depth. A 500 Mb SNP backbone panel (Agilent Technologies) was added to assist with gene amplification/deletion measurements. HPV16/18 was detected using the Caris pipeline, which includes 39 unique baits to detect HPV16 and 50 unique baits to detect HPV18 out of a total of 2360 total pathogen baits. The threshold for positive is ≥100 reads for either HPV16 or HPV18.

### 2.5. Whole Transcriptome Sequencing and Immune Cell Infiltration

Qiagen RNA FFPE tissue extraction kit was used for extraction, and the RNA quality and quantity were determined using the Agilent TapeStation. Biotinylated RNA baits were hybridized to the synthesized and purified cDNA targets, and the bait–target complexes were amplified in a post-capture PCR reaction. The Illumina NovaSeq 6500 was used to sequence the whole transcriptome from patients to an average of 60 M reads. Raw data were demultiplexed by the Illumina Dragen BioIT accelerator, trimmed, counted, removed of PCR-duplicates, and aligned to human reference genome hg19 by the STAR aligner. For transcription counting, transcripts per million molecules were generated using the Salmon expression pipeline. Immune cell fraction was calculated by Quantiseq using transcriptome data [27].

### 2.6. Survival Analysis

Real-world overall survival (rwOS) information was obtained from insurance claims data, and Kaplan–Meier estimates were calculated from the first date of contact to the last date of contact or the first day of treatment to the last day of treatment (TOT).

### 2.7. Statistics

Statistical significance was determined using the Chi-Squared test and Benjamini–Hochberg correction for multiple comparisons. Kaplan–Meier estimates were calculated for molecularly defined patient cohorts. Significance was determined as *p* values <0.05.

## 3. Results

### 3.1. Patient Characteristics

A total of 2905 HNSCC patients were identified in the Caris database, of which 948 were OPSCC. Ages ranged from 15 to 90 years, and the median age of the cohort was 68 years (Table 1). Smoking status was available for 525 patients, and 41% of patients (215/525) were smokers. Among those who were tested for p16 and/or HPV, 32% (251/791) expressed p16 and 28% (91/326) were HPV+. The majority of p16+ tumors were OP in origin (68%, 171/251). In the OPSCC group, 41% were p16+ (171/420) and 52% were HPV positive (71/148).

### 3.2. Genomic and Molecular Landscape between p16+ and p16−

The most common mutation in the entire cohort of HNSCC was *TP53* (54%)*,* followed by *CDKN21* (17%*). TP53* (33%), *PIK3CA* (17%), and *KMT2D* (10.7%) were the most common mutations identified in OPSCC (Table 2). The *TP53* mutation was predominant in p16− OPSCC (49%) and non-OP HNSCC (58%) tumors in contrast to p16+ OPSCC (10%) (*p* < 0.0005). *PIK3CA* and *KMT2D* were the most common mutations in p16+ OPSCC (Figure 1A), while *TP53* and *TERT* mutations were the most common in non-OP HNSCC regardless of p16 status (Figure 1B–D). *NOTCH1, CDKN2A,* and *TERT* mutations were more prevalent in OPSCC tumors that were p16− or HPV− in contrast to OPSCC tumors that were p16+ or HPV+ (*p* < 0.05) (Figure 1E). When the entire cohort of HNSCC patients were analyzed (Figure 1F), there were discrepancies in commonalities of mutation frequencies between p16+ and HPV+ groups. *Rb* was more frequently mutated in p16+ compared to p16− but not detected in the HPV+ group, and *KRAS* was more likely mutated in HPV+ with lower rates in p16+ groups and not detected in p16− or HPV− groups. The most frequently identified hotspot *TP53* mutations were in codons G245A, R248W, R248Q, G245F, and R248G in p16+ OPSCC and R175H, R248W, R273C, H179Y, and R273L in p16+ non-OP HNSCC. *FGF3, CCND1, FGF4,* and *FGF19* copy number alterations (CNA) were less common in p16+ OPSCC when compared to p16− or HPV16− OPSCC (*p* < 0.0005) (Figure 2).

### 3.3. ICB Biomarker Comparison in p16+ vs. p16− OPSCC

Several markers have been used to predict the response to ICB, including tumor mutational burden (TMB), microsatellite instability (MSI), programmed death-ligand 1 (PD-L1) expression, and tumor immune cell infiltration. PD-L1 positivity was 87%, and 16% had TMB (≥10Mb) for the entire HNSCC cohort. No statistical difference was detected in TMB (≥10Mb), MSI, or PD-L1 between the p16 and HPV OPSCC and non-OP HNSCC groups (Figure 3 and data not shown). However, B-cell, myeloid dendritic cells, and NK cell infiltration was enriched in p16+ versus p16− OPSCC, and neutrophil presence was reduced in p16+ tumors (Figure 4). Conversely, there was no statistically significant difference in macrophage (M1 and M2) and CD8+ T cells between the subgroups.

### 3.4. Survival Outcomes in p16+ and p16− Disease

Similar to previous reports, p16+ OPSCC patients had a longer survival rate compared to p16− patients with rwOS of 33 vs. 19 months (HR = 0.597, *p* = 0.001), respectively (Table 3 and Figure 5B). However, there was no difference in time on treatment (TOT) (4.2 vs. 2.8 months, HR 0.796, *p* = 0.221) between p16+ and p16− OPSCC groups treated with ICB, respectively (Table 3 and Figure 5C). For the non-OP HNSCC cohort, we also detected a longer rwOS for the p16+ group compared to the p16− group similar to OPSCC (34 vs. 17 months, HR 0.551, *p* = 0.0001, Table 3 and Figure 6B). Converse to the OPSCC group, when non-OP HNSCCs were stratified by treatment with ICB, TOT was higher in the p16+ group compared to the p16− group treated with ICB (4.3 vs. 3.3 months, HR 0.632, *p* = 0.016, Table 3 and Figure 6C).

## 4. Discussion

Our findings are consistent with previous work and confirm that *TP53*, *NOTCH1*, *CDKN2A*, *TERT,* and *PIK3CA* are the most frequent mutations in OPSCC [28,29]. Several of these mutations are under investigation as possible therapeutic targets. PI3K inhibitors such as BKM120 or BYL719 have been investigated alone or in combination with other agents in multiple cancers, including HNSCC [30]. However, it remains unclear if these mutations serve as independent drivers of pathogenesis and predictors of survival, necessitating further validation and pathway analysis. 

The frequency of HPV+ (52%) and P16+ (41%) in the OPSCC group was lower than previously reported (~70%) in the literature [31]. However, this result could be skewed given that the majority of our cohort likely included patients who had relapsed or had metastatic HNSCC. We noted some OPSCC patients with discordant p16 and HPV status (13 p16+/HPV− and 14 p16−/HPV+ out of 125 cases (Appendix A in Appendix A)), accounting for about 22% of OPSCC cases. Lewis et al. has demonstrated that p16 serves as a superior predictor compared to HPV detection for risk stratification of OPSCC [6]. Comparison of outcomes between OPSCC patients that were p16+ and HPV+ versus those that were p16+ but HPV− demonstrated no difference in survival in the study [6]. Supporting p16 as a strong predictor of prognosis [32], we also detected better survival in our p16+ non-OPSCC group, whose members were more commonly HPV negative. Therefore, p16 remains a commonly used marker in most centers for risk stratification but understanding the discordance may be relevant in larger populations. 

The advent of ICB has revolutionized the treatment paradigm of R/M HNSCC. Immune markers such as PD-L1 and tumor mutational burden (TMB) have emerged as predictors of immune response in various clinical trials [16,17,19,23,33]. Our data correlate with prior reports of PD-L1 positivity of ~85–98% in OPSCC [24,25]. We found no difference in PD-L1 staining in p16+ and p16− OPSCC. Therefore, PD-L1 as a biomarker and ICB response predictor is less impactful in this group. HPV (p16)-related carcinogenesis has been linked to lower rates of TMB but higher frequency of epigenetic changes leading to oligoclonal tumors that have a higher sensitivity to chemotherapy and radiation as well as ICB [34,35]. In our study, we identified no difference in rates of TMB between p16 groups, but this may be due to the small sample size of patients harboring *TMB ≥ 10/Mb* (10%).

*TP53* mutations were more common in p16− (49%) tumors in contrast to p16+ (10%) (*p* < 0.0005), which is concordant with what was reported by other studies [34]. The prevalence of hotspot *TP53* mutations was similar to that previously reported, including *TP53* missense mutations at codons R248, R273, G245, R175, R282, and H179 as the most common hotspot mutations in HNSCCs [36]. *TP53* mutation has been repeatedly linked to poor outcomes in various malignancies [30], along with low response to ICB [37]. Targeting the mutation has been proposed by many to offset the poor responses to treatment, including ICB. For example, WEE1 kinase inhibitor adavosertib (AZD1775) has shown benefits in TP53 mutant HNSCC [38]. Further studies are under investigation.

Our data concur with previous reports that p16+ OPSCC patients have superior OS compared to p16− patients. However, in the OPSCC p16+ and p16− groups who received ICB, there was no statistically significant difference in the rwOS or TOT [10]. These observations could be due to good outcomes in OPSCC patients regardless of the treatment type, short follow-up and/or small sample size, or possibly p16 directed alterations independent of HPV infection.

In contrast, TOT for p16+ non-OP HNSCC patients receiving ICB was longer compared to p16− patients; this finding was not reproduced in the OPSCC subgroup. While some studies have observed longer survival in p16/HPV+ non-OP HNSCC [39], the outcomes of ICB in p16+ non-OP HNSCC have not been validated in the literature. Notably, a few studies have reported that the survival advantage of p16/HPV does not extend to the non-OP HNSCC [40]. However, many of these studies had smaller numbers, whereas ours is one of the largest cohorts reported. In addition, our cohort involved mostly R/M HNSCC, while previous reports may have had a combination of local advanced and R/M cases. Larger cohorts studied prospectively would be required to elucidate possible genomic/molecular factors in this rare p16+ non-OP HNSCC subgroup. In addition, randomized controlled trials are required to verify the significance of p16 as a prognostic marker for ICB therapy in both OPSCC and non-OP HNSCC.

Limitations of our study include its retrospective nature, the lack of subjects’ descriptive oncology history because the data were extracted from insurance claims, and the paucity of treatment information around cases prior to obtaining tissue; patients likely received heterogeneous treatment prior to the current data analysis.

## 5. Conclusion

The molecular and genetic profiling of cancers may enlighten new biomarkers of response as well as potential therapeutic targets. Here, we confirm previous findings that p16+ HNSCC patients have improved survival compared to those with p16− HNSCC. Although we did not detect improved survival in p16+ OPSCC patients upon treatment with ICB, interestingly, p16+ non-OP HNSCC had longer TOT, suggesting improved response to ICB compared to those with p16− disease. In the future, these results may help guide treatment decisions and provide a rationale for further investigation. Clinical trials with large patient populations are required to assess whether p16 and other potential biomarkers can predict ICB treatment response.

## Figures and Tables

**Figure 1 cancers-13-06309-f001:**
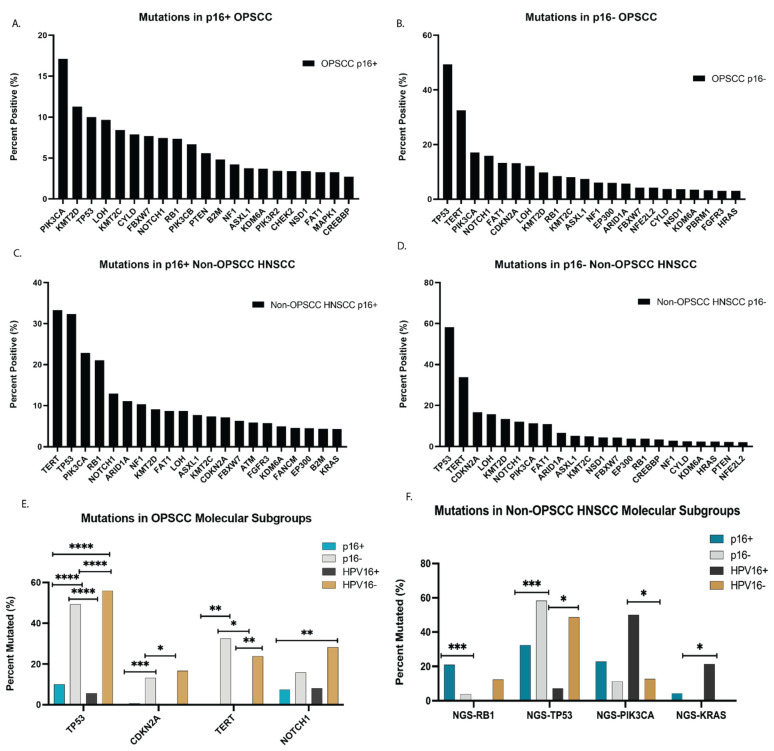
Genetic landscape of p16+ and p16− OPSCC. (**A**–**D**) Whole-exome sequencing (WES), targeted Next-Generation Sequencing (NGS), and whole-transcriptome sequencing (WTS) were performed to identify the most common mutations and are graphically represented by labeled subgroups. (**E**,**F**) OPSCC and HNSCC groups were analyzed to detect the most prevalent mutations in p16 and HPV positive and negative groups. Statistical significance was determined using the chi-squared test and Benjamini–Hochberg correction. **** denotes *p* < 0.00005, *** denotes *p* < 0.0005, ** denotes *p* < 0.005, * denotes *p* < 0.05.

**Figure 2 cancers-13-06309-f002:**
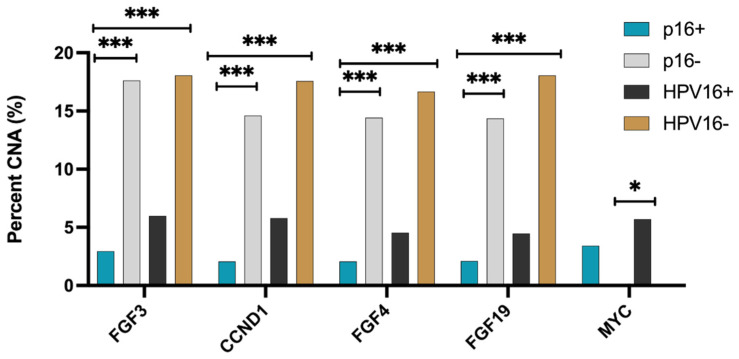
Copy Number Alterations (CNA) in p16+/− and HPV16 +/− OPSCC. Copy number amplifications were detected by WES in p16+, p16−, HPV16+, HPV16− oropharyngeal tumors. Statistical significance was determined using the chi-squared test and Benjamini–Hochberg correction. *** denotes *p* < 0.0005 * denotes *p* < 0.05.

**Figure 3 cancers-13-06309-f003:**
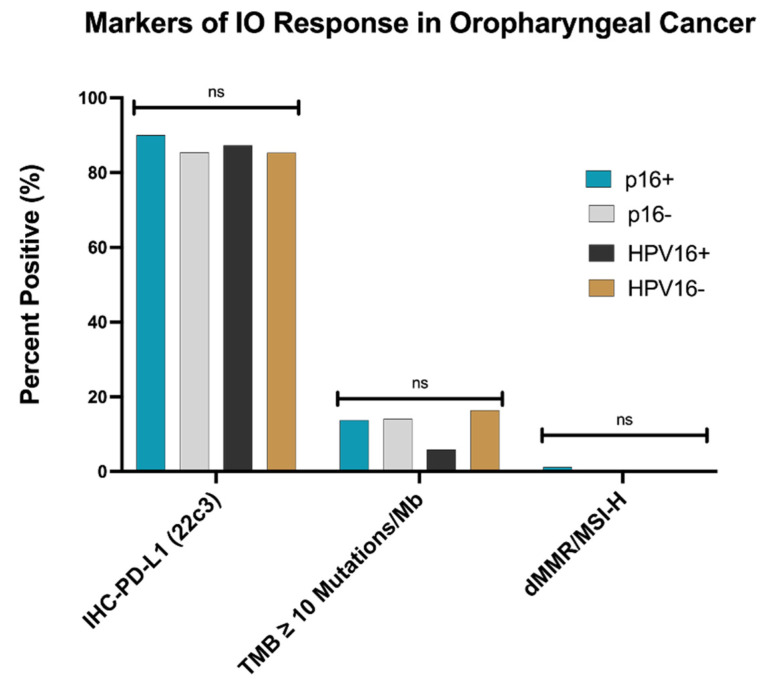
Markers of ICB response in OPSCC. Markers of immune checkpoint inhibitor response (PD-L1 (CPS ≥ 1), TMB ≥ 10/Mb, and dMMR/MSI-H status) were measured in p16+, p16−, HPV16+, and HPV− oropharyngeal cancers. Statistical significance was determined using the chi-squared test and Benjamini–Hochberg correction. No significance (ns) was defined as *p* > 0.05.

**Figure 4 cancers-13-06309-f004:**
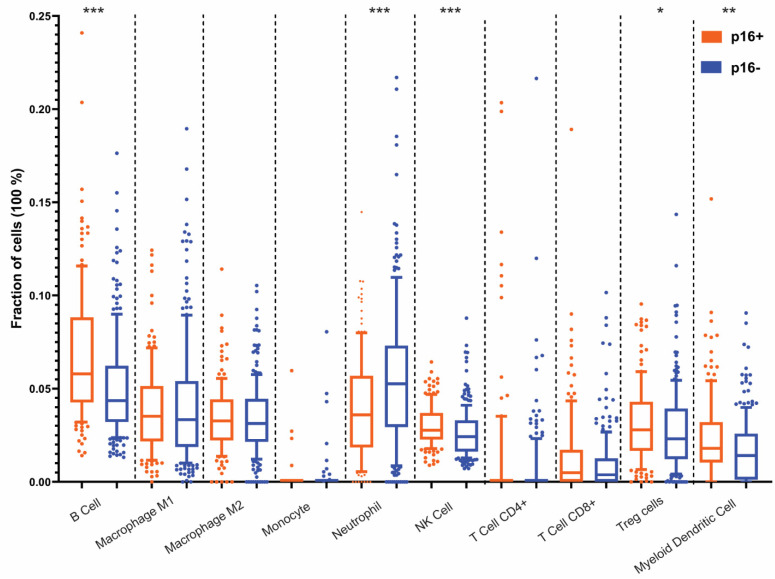
Immune cell infiltration in p16+ vs. p16− OPSCC. Immune cell fractions were calculated using QuanTIseq computational pipeline and RNA-seq data. The following immune cells were assessed: B cells, M1 and M2 macrophages, Monocytes, Neutrophils, NK cells, CD4+ and CD8+ T cells, Treg cells, and Myeloid dendritic cells in p16+ and p16−oropharengeal tumors. Statistical significance was determined using the chi-squared test and Benjamini–Hochberg correction. *** denotes *p* < 0.0005, ** denotes *p* < 0.005, * denotes *p* < 0.05.

**Figure 5 cancers-13-06309-f005:**
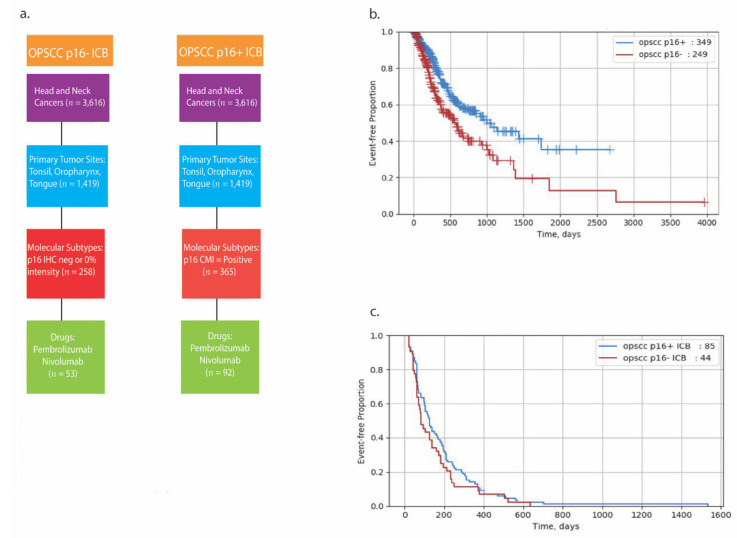
Real-World Overall Survival (rwOS) in OPSCC patients. Consort diagram detailing real-world data cohorts (**a**) Kaplan–Meier curves representing (**b**) rwOS in p16+ vs. p16− OPSCC and (**c**) TOT in p16+ vs. p16− OPSCC treated with ICB.

**Figure 6 cancers-13-06309-f006:**
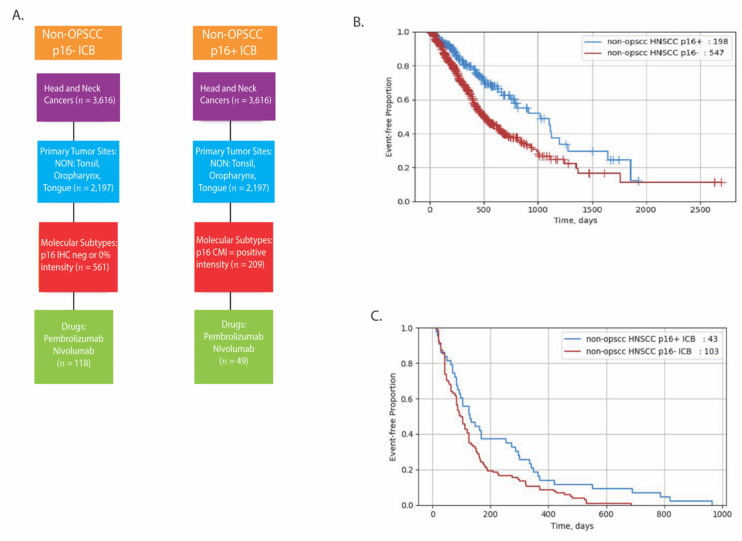
Real-World Overall Survival in non-OP HNSCC patients. Consort diagram detailing real-world data cohorts (**A**). rwOS in non-OP HNSCC. Kaplan–Meier curves representing (**B**) rwOS in p16+ vs. p16− non-OP HNSCC and (**C**) TOT in p16+ vs. p16− non-OP HNSCC treated with ICB.

**Table 1 cancers-13-06309-t001:** Demographics.

Age (Years)	Median—68 (Range 15–90)
Gender	Male 76.4% (2219/2905) Female 23.6% (686/2905)
Smokers	41 % (215/525)
P16+	32% (251/791)
HPV+	28% (91/326)
P16+ and HPV+ OPSCC	38% (51/134)
P16+ and HPV+ Non-OPSCC HNSCC	5% (10/182)
Primary	57% (1646/2905)
Recurrent/Metastatic	43% (1259/2905)

**Table 2 cancers-13-06309-t002:** Molecular profiling of p16+ vs. p16− OPSCC.

Molecular Features	All OPSCC	All Non-OP HNSCC	OPSCC p16+	Non-OP HNSCC p16+	OPSCC p16−	Non-OP HNSCC p16−
**PD-L1 ≥ 1 (22c3)**	86.88% (342/394)	87.59% (628/717)	90% (154/171)	86.07% (68/79)	85% (211/247)	91.94% (502/546)
* **TP53** *	33% (227/686)	63.88% (888/1390)	10% (14/140)	32.36% (22/68)	49% (108/219)	58.33% (280/480)
**TMB ≥ 10/Mb**	10% (48/463)	18.69%’ (168/899)	13% (20/145)	25.76% (17/66)	14.1% (30/213)	16.88% (81/480)
* **NOTCH1** *	9.2% (59/654)	10.48% (138/1318)	7.4% (9/121)	12.9% (8/62)	15.9% (30/189)	12.05% (50/415)
* **CDKN2A** *	7.6% (44/576)	22.09% (237/1073)	0.6% (1/148)	7.14% (5/70)	13.2% (28/212)	16.60% (78/470)
* **TERT** *	3.4% (10/291)	8.38% (45/537)	0% (0/28)	33.33% (3/9)	32.5% (13/40)	33.67% (33/98)
* **PIK3CA** *	17.1% (120/702)	11.31% (159/1405)	17.1% (25/146)	22.86% (16/70)	17.1% (38/222)	11.25% (55/489)
* **KMT2D** *	10.7% (61/572)	12.69% (139/1095)	11.2% (16/142)	9.09% (6/66)	9.8% (21/214)	13.38% (63/471)

**Table 3 cancers-13-06309-t003:** rwOS and TOT (months) in p16+ vs. p16− OPSCC and non-OP HNSCC cohorts.

rwOS and TOT	P16+ (Months)	P16− (Months)	HR	*p* Value
**Non-OP HNSCC (rwOS)**	34	17	0.551	0.0001
**OPSCC (rwOS)**	33.3	19.1	0.597	0.001
**Non-OP HNSCC treated with ICB (TOT)**	4.3	3.3	0.632	0.016
**OPSCC treated with ICB (TOT)**	4.2	2.8	0.796	0.221

## Data Availability

No publicly archived datasets were used for this study.

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
