# Peer review of "Genomic and Molecular Profiling of Human Papillomavirus Associated Head and Neck Squamous Cell Carcinoma Treated with Immune Checkpoint Blockade Compared to Survival Outcomes"

_cancers, 2021, doi:10.3390/cancers13246309_

Round 1

Reviewer 1 Report

The paper by Sheikh et al. investigates the association between genomic and molecular alterations and response to immunomodulatory therapy estimated as survival in head and neck carcinomas with a focus on oropharyngeal cancers, mainly HPV+. The paper investigates a very interesting topic and aims at finding predictive markers. Many of the findings reported by the Authors corroborated previously reported findings. The important novelty is the analysis of data in the context of ICB therapy.

The paper is very well prepared. The study hypothesis is very clearly stated. Methods are described very concisely. The results are presented in a clear way and the discussion and conclusions are supported by the results.

There are only minor remarks:

  • affiliations - please use the same format for all Authors, including country of origin
  • delete the dot at the end of the title (line 5)
  • line 17 - add "patients" after the bracket
  • line 25 - add "patients" after the bracket
  • line 42 - I would suggest deleting "treatment" and "genome" from keywords (too general)
  • line 63 - should be "PIK3CA" (italics)
  • line 64 - should be "TP53" instead of p53
  • line 68 - please, explain the abbreviation "ORR"
  • line 163 - please, explain the abbreviation "IRB"
  • Table 1 - it would good to include TNM data and the number of primary and recurrent and metastatic cases
  • line 220 - delete "in" (repeated twice)

Author Response

  • affiliations - please use the same format for all Authors, including country of origin - done
  • delete the dot at the end of the title (line 5) - done
  • line 17 - add "patients" after the bracket - done
  • line 25 - add "patients" after the bracket - done
  • line 42 - I would suggest deleting "treatment" and "genome" from keywords (too general) - done
  • line 63 - should be "PIK3CA" (italics) - done
  • line 64 - should be "TP53" instead of p53 - done
  • line 68 - please, explain the abbreviation "ORR" - done
  • line 163 - please, explain the abbreviation "IRB" - done
  • Table 1 - it would good to include TNM data and the number of primary and recurrent and metastatic cases
  • line 220 - delete "in" (repeated twice) - done

Reviewer 2 Report

In their study, the authors set themselves the ambitious task of analyzing the impact of p16/HPV of status HNSCC on response to immune checkpoint blockade and overall survival. The article is written clearly enough. Despite extensive analyzes with genomic and molecular profiling of tumors, however, due to the retroactive nature of the study and the lack of primarily clinical patient data (patients’ data was extracted from insurance claims), the conclusions of the study are sparse. Above all, it is disturbing that the authors do not state a clear conclusion that would summarize the results of the analyzes performed (in the Abstract as well as in the Discussion).

Minor comments:

Introduction, L61. »The same was true in recurrent/metastatic (R/M) HNSCC.« - This statement must be supported by the reference.

L70: ORR – abbreviation must be properly introduced

Results/Patients characteristics: Add the information on the patients with a p16 and HPV test and what proportion was p16 and HPV positive in the overall HNSCC population and among OPSCC patients?

Results/Survival outcomes: Information on the rwOS is in Table 3! (but not in Table 2)

Figure 4: Add an explanation of what the stars at the top of the graph mean

Supplementary material, Discussion pg 10/l256-257: according to Fig S1, among oropharyngeal SCCs the number of p16/HPV discordant cases is at least 25! This is not in agreement with the statement that »We noted a minority with discordant p16 and HPV status (7 p16+/HPV-313, 8 p16-/HPV+) among 248 patients who had both p16 and HPV status available«.

Discussion, pg.10, paragraph 1: I would suggest adding the ref.: Chung CH et al. p16 protein expression and human papillomavirus status as prognostic biomarkers of nonoropharyngeal head and neck squamous cell carcinoma. J Clin Oncol. 2014 Dec 10;32(35):3930-8. doi: 10.1200/JCO.2013.54.5228.

Author Response

In their study, the authors set themselves the ambitious task of analyzing the impact of p16/HPV of status HNSCC on response to immune checkpoint blockade and overall survival. The article is written clearly enough. Despite extensive analyzes with genomic and molecular profiling of tumors, however, due to the retroactive nature of the study and the lack of primarily clinical patient data (patients’ data was extracted from insurance claims), the conclusions of the study are sparse. Above all, it is disturbing that the authors do not state a clear conclusion that would summarize the results of the analyzes performed (in the Abstract as well as in the Discussion).

Response: Thank you for reviewing our manuscript and sharing your valuable opinions. We have better summarized conclusions in the manuscript.

Minor comments:

Introduction, L61. »The same was true in recurrent/metastatic (R/M) HNSCC.« - This statement must be supported by the reference.

Response: Following references have been included to support the above-mentioned statement.

Fleming, C.W.; Ward, M.C.; Woody, N.M.; Joshi, N.P.; Greskovich, J.F.; Rybicki, L.; Xiong, D.; Contrera, K.; Chute, D.J.; Milas, Z.L.; et al. Identifying an oligometastatic phenotype in HPV-associated oropharyngeal squamous cell cancer: Implications for clinical trial design. Oral Oncology 2021, 112, 105046, doi:https://doi.org/10.1016/j.oraloncology.2020.105046.

Fakhry, C.; Zhang, Q.; Nguyen-Tan, P.F.; Rosenthal, D.; El-Naggar, A.; Garden, A.S.; Soulieres, D.; Trotti, A.; Avizonis, V.; Ridge, J.A.; et al. Human papillomavirus and overall survival after progression of oropharyngeal squamous cell carcinoma. J Clin Oncol 2014, 32, 3365-3373, doi:10.1200/jco.2014.55.1937.

Faraji, F.; Eisele, D.W.; Fakhry, C. Emerging insights into recurrent and metastatic human papillomavirus-related oropharyngeal squamous cell carcinoma. Laryngoscope Investig Otolaryngol 2017, 2, 10-18, doi:10.1002/lio2.37.

L70: ORR – abbreviation must be properly introduced

Response: Abbreviation ORR has been explained

Results/Patients characteristics: Add the information on the patients with a p16 and HPV test and what proportion was p16 and HPV positive in the overall HNSCC population and among OPSCC patients?

Response: Suggested change has been made in the manuscript.

Results/Survival outcomes: Information on the rwOS is in Table 3! (but not in Table 2)

Response: The referenced error has been corrected.

Figure 4: Add an explanation of what the stars at the top of the graph mean

Response: Suggested change has been made in the manuscript.

Supplementary material, Discussion pg 10/l256-257: according to Fig S1, among oropharyngeal SCCs the number of p16/HPV discordant cases is at least 25! This is not in agreement with the statement that »We noted a minority with discordant p16 and HPV status (7 p16+/HPV-313, 8 p16-/HPV+) among 248 patients who had both p16 and HPV status available«.

Response: We thank the reviewer for indicating this error. We have now corrected this. The total number of p16/HPV discordant cases in our data was 27 (13 p16+/HPV-, 14 p16-/HPV+). Given this is 22%, we agree that this is a bit high if only taking into account OPSCC but only about 15% of cases were discordant overall. We have updated the manuscript.  

Discussion, pg.10, paragraph 1: I would suggest adding the ref.: Chung CH et al. p16 protein expression and human papillomavirus status as prognostic biomarkers of nonoropharyngeal head and neck squamous cell carcinoma. J Clin Oncol. 2014 Dec 10;32(35):3930-8. doi: 10.1200/JCO.2013.54.5228. – done

Response: Suggested reference by the reviewer has been included in the manuscript.
